# Divergent regioselective Heck-type reaction of unactivated alkenes and *N*-fluoro-sulfonamides

Chunyang Zhao[1,4], Yang Li[1,4], Yujiao Dong[2,4], Miao Li[1], Dan Xia[1], Shuangqiu Gao[1], Qian Zhang [1], Qun Liu[1], Wei Guan [2] ✉ & Junkai Fu [1,3] ✉

The control of regioselectivity in Heck-type reaction of unactivated alkenes represents a longstanding challenge due to several detachable hydrogens in β−H elimination step, which generally afford either one specific regioisomer or a mixture. Herein, a copper-catalyzed intermolecular Heck-type reaction of unactivated alkenes and *N*-fluoro-sulfonamides with divergent regioselectivities is reported. The complete switch of regioselectivity mainly depends on the choice of different additives. Employment of alcohol solvent gives access to vinyl products, while the addition of carboxylate leads to the formation of allylic products. In addition, exclusion of these two promoting factors results in β-lactams via a C−N reductive elimination. This protocol shows a broad substrate scope for both alkenes and structurally diverse *N*-fluoro-sulfonamides, producing the corresponding products with excellent regio- and stereoselectivities. Further control experiments and DFT calculations provide in-depth insights into the reaction mechanism, highlighting the distinct effect of the additives on a bidentate auxiliary-stabilized Cu(III) intermediate.

The intermolecular Mizoroki–Heck reaction is arguably one of the most significant methods for the construction of C–C bonds[1–6]. Electronically biased alkenes, such as styrene and acrylate derivatives are generally required to achieve both high reaction efficiency and excellent regioselectivity. In comparison, the coupling of unactivated alkenes has been far less investigated due to the intrinsic low reactivity of these olefins and the difficulty in controlling the linear regioselectivity[7,8]; the latter is attributed to the existence of several detachable β-hydrogens ($H_a$ vs $H_b$) as well as the interconversion between vinyl and allylic products in the presence of fleeting metal hydride species (Fig. 1a)[9,10]. In the past few decades, unremitting efforts have been devoted to this field[11,12]. White et al. pioneered a substrate-chelated oxidative Heck reaction of phenylboronic acids and unactivated alkenes tethered with proximal heteroatoms for the regioselective synthesis of vinyl products[13–18]. Shortly thereafter, the Sigman group reported a catalyst-controlled β-H

elimination process in both oxidative and classic Heck reactions of unactivated alkenes, giving good selectivity for the *E*-styrenyl products. But a mixture of vinyl and allylic products was obtained for γ-functionalized alkenes (R = $CO_2$Me or Ph), and the ratios mainly depended on the electronic effect of the substituents on the aryl groups[19–21]. A significantly different Heck-type reaction was reported by the Gaunt group, wherein diaryliodonium salts coupled with unactivated alkenes via a carbocation intermediate, favoring unusual allylic products with moderate to good regio- and stereoselectivities[22]. Compared to aryl components, the intermolecular coupling of alkyl electrophiles and unactivated alkenes was previously regarded as an insurmountable obstacle[8,23], considering the sluggish oxidative addition of low-valent transition metal to C(sp³) electrophiles and the facile β-H elimination[24]. This hurdle was overcome until very recently via electrophilic alkyl radicals stabilized by adjacent functionalities (e.g., F,

[1]Jilin Province Key Laboratory of Organic Functional Molecular Design & Synthesis, Department of Chemistry, Northeast Normal University, Changchun 130024, China. [2]Institute of Functional Material Chemistry, Department of Chemistry, Northeast Normal University, Changchun 130024, China. [3]State Key Laboratory of Chemical Oncogenomics and Key Laboratory of Chemical Genomics, Peking University Shenzhen Graduate School, Shenzhen 518055, China. [4]These authors contributed equally: Chunyang Zhao, Yang Li, Yujiao Dong. ✉e-mail: guanw580@nenu.edu.cn; fujk109@nenu.edu.cn

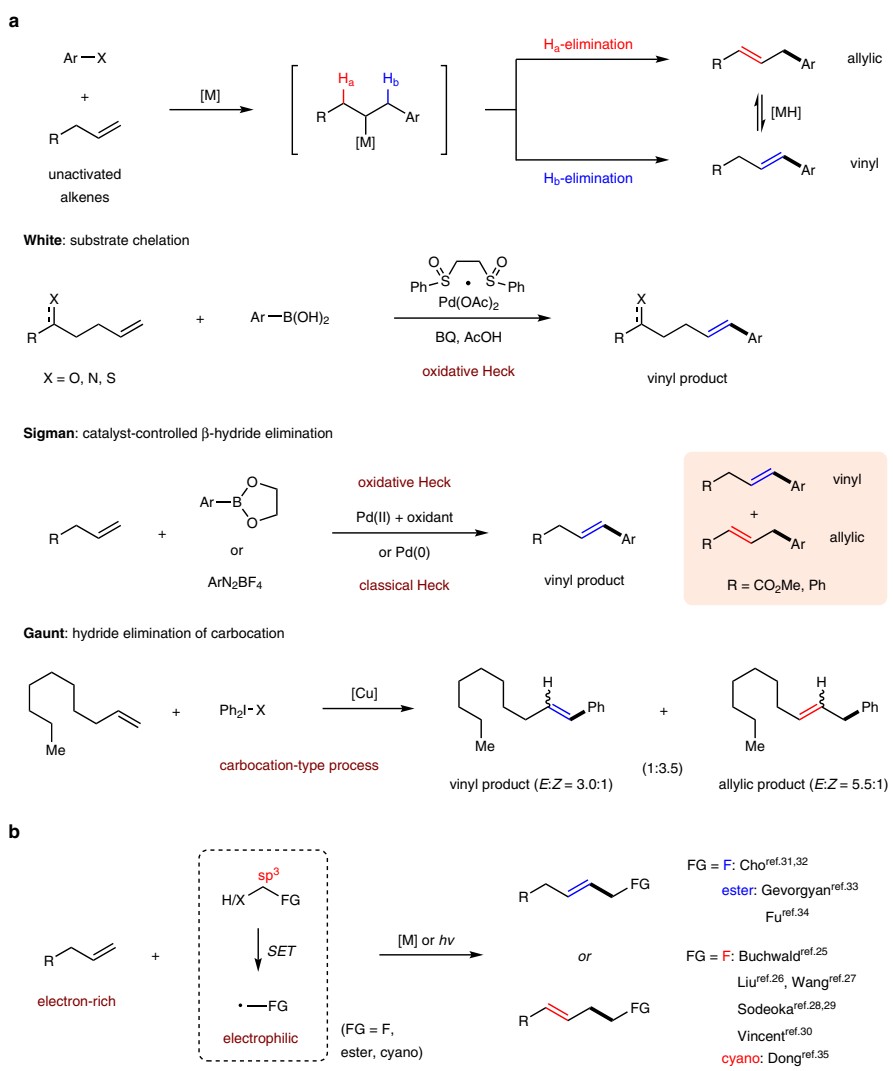

**Fig. 1 | The regioselectivity competition for intermolecular Heck-type reaction of unactivated alkenes. a** The regioselectivity in Heck reaction with aryl components. **b** The regioselectivity in Heck-type reaction with alkyl electrophiles. BQ benzoquinone.

ester, or cyano), pioneered by Buchwald[25], Liu[26], Wang[27], Sodeoka[28,29], Vincent[30], Cho[31,32], Gevorgyan[33], Fu[34], and Dong[35]. However, the lack of benzylic hydrogens renders the control of regioselectivity more challenging, and the preference for vinyl or allylic products heavily relies on different reaction partners (Fig. 1b). Despite the aforementioned encouraging advancements, only one specific regioisomer or a mixture of isomers are obtained in all successful cases; in contrast, a switch of regioselectivity in a single Heck operation of unactivated alkenes by simply adjusting the reaction conditions has not yet been realized.

Our group has long sought alkyl electrophiles for the intermolecular Heck-type reactions of unactivated alkenes. In order to extend the scope of this reaction to commonly encountered nucleophilic alkyl radicals without the stabilization by an adjacent functionality, a bidentate directing group[36–39] was employed in combination with copper salt to reduce π electron density of the electron-rich olefin moiety, making the polarity mismatched coupling of nucleophilic alkyl radicals with unactivated alkenes a reality[40], and the vinyl products were obtained via concerted β−$H_b$ elimination of the key Cu(III) intermediate **A**[34,41]. Nevertheless, a regioselectivity switch to $H_a$ elimination remains a challenge (Fig. 2). Situated differently than the rotatable $H_b$, the location of $H_a$ in a metallacyclic intermediate with locked conformation impedes the $H_a$ elimination, both sterically and electronically[42]. Moreover, as a highly reactive species, the Cu(III)

complex **A** is easy to undergo a reductive elimination to form C−N bond accompanying with the hydride elimination process[43].

In recent years, the alkyl radicals generated from N-fluoro-amides via a sequence involving single-electron transfer (SET) with transition metal or photocatalyst, followed by 1,5-hydrogen atom transfer (HAT) have been well studied in the construction of C−C and C−X bonds[44–53]. However, the interaction of N-fluoro-amides with alkenes has rarely been explored. A seminal work was disclosed very recently by Li in benzylic radical-initiated difunctionalization of styrene derivatives[54].

Herein, we report a copper-catalyzed intermolecular Heck-type reaction of unactivated alkenes and N-fluoro-sulfonamides with divergent regioselectivities by simply varying external additives. When the reactions were carried out in alcohol solvent, vinyl products **B** were selectively obtained via a hydroxyl-assisted concerted β−$H_b$ elimination; while a complete regioselectivity switch to concerted β−$H_a$ elimination (E2-type elimination) was achieved under acidic conditions in the presence of carboxylate, which accelerated the deprotonation of α-C−H through an amide-directed eight-membered metallacyclic transition state to deliver allylic products **C**. In addition, without these promoting factors, alternative β-lactams **D** could be formed through a reductive elimination of the Cu(III) intermediate **A**. Collectively, this protocol represented an actionable roadmap toward a broadly accessible solution to the assembly of branches of Heck-type reaction entities, with

**Fig. 2 | Divergent regioselective Heck-type reaction.** External additives controlled divergent regioselectivities in intermolecular Heck-type reaction of unactivated alkenes and *N*-fluoro-sulfonamides (this work).

broad substrate scope for both terminal and internal alkenes, good *E*-stereoselectivity, and excellent functional-group tolerance.

## Results

### Screening of the reaction conditions

The study was initiated using 8-aminoquinoline (AQ) amide-tethered[55] alkene **1a** and *N*-fluoro-sulfonamides **2a** bearing a δ-tertiary carbon as model substrates in the presence of Cu(TFA)$_2$·H$_2$O (Table 1). Screening began with an investigation of reaction solvent. For nonpolar solvent (entries 1–2), the reaction performed in DCE afforded reductive elimination product **5a** in 42% yield, while in toluene a mixture of coupling products was obtained in 16% yield with allylic product **4a** as the major isomer (*r.r.* = 1:3). Switching to polar solvent dioxane led to a significant boost in reductive elimination, providing β-lactam **5a** in 73% (entry 3, as conditions C). However, CH$_3$CN drastically attenuated the reaction profile, with the primary mass balance being unreacted starting material **1a** and minor amounts of reductive elimination product **5a** (entry 4). Delightedly, the reaction in NMP gave coupling products **3a** and **4a** as the major products with a good *r.r.* value of 10:1 (entry 5). The employment of alcohol as reaction solvent further benefitted the reaction efficiency and regioselectivity; in ^*i*^BuOH, the vinyl product **3a** was afforded in 65% yield in a highly regio- (*r.r.* > 20:1) and stereoselective (*E/Z* > 20:1) manner, and the reductive elimination process could be totally suppressed (entries 6–7). Then other copper salts were tested, and the results showed that Cu(OAc)$_2$·H$_2$O was the best choice to produce **3a** in 72% yield, while Cu(NO$_3$)$_2$ proved to be completely ineffective (entries 8–11). Considering the important role of solvent for this reaction, some mixed solvents were evaluated (entries 12–14), among which an ^*i*^BuOH/CH$_3$CN combination could improve the yield to 82%, maintaining high regio- and *E*-stereoselectivities (*r.r.* > 20:1, *E/Z* > 20:1) (entry 14, as conditions A). The necessity of the copper catalyst for this transformation was demonstrated via a control experiment with the complete exclusion of copper salt, which only recovered the starting material **1a** (entry 15). Other types of transition metals have also been tested, such as Ni(cod)$_2$ and Co(acac)$_2$, but no desired product could be detected (entries 16–17).

During the screening process, an intriguing result was observed when the reaction was conducted in toluene (entry 2). Despite low yield, the domination of **4a** in coupling product distribution (*r.r.* = 1:3) meant the possibility of a regioselectivity switch to allylic products. Encouraged by this hypothesis, more nonpolar solvents were tested (entries 18–19), and the *r.r.* value could be elevated to 1:8 in benzene with a low overall yield of 15% for the coupling products. To improve the reaction efficiency, we envisaged that the weak α-acidity of the amide group might offer a handle to base-promoted E2-type elimination. In this context, we attempted the addition of a base to facilitate the concerted β–H$_a$ elimination (entries 20–21). However, both the coupling and reductive elimination processes were completely suppressed; only the alkene substrate isomerization could be observed, and the α,β-unsaturated amide **1a′** was isolated in 90% yield in the presence of DBU. After an extensive screening, to our delight, metal carboxylates were found to be crucial to improve both the yield and *r.r.* value of the Heck-type reaction (entries 22–24). When the reaction was carried out in the presence of 0.4 equiv of Zn(OAc)$_2$, allylic product **4a** was obtained in 45% yield as a single regioisomer (*r.r.* > 20:1) along with the formation of β-lactam **5a** in 32% yield. Increasing the loading of Zn(OAc)$_2$ to 1.0 equiv partly suppressed the formation of β-lactam **5a** (15%); however, the yield of the desired coupling product **4a** (35%) did not increase accordingly and 28% of the alkene substrate was recovered (entry 25). The increased loading of Zn(OAc)$_2$ accelerated the decomposition of *N*-fluoro-sulfonamide **2a** (observed by TLC), which resulted in a decreased yield and a recovery of the alkene substrate. We also tested some other zinc salts, such as ZnBr$_2$·TMEDA that has been used as an additive by Nakamura in iron-catalyzed C–H bond functionalization[56], but it made little effect on the outcome (entry 26), showing a significant influence of the carboxylate on the reaction[57]. Further screening of the copper salts including Cu(OAc)$_2$·H$_2$O and Cu(CH$_3$CN)$_4$PF$_6$ gave inferior yields of **4a**, while alkene substrate **1a** was unresponsive in the presence of other reductive metal catalyst[44], e.g., Fe(OTf)$_2$ (entries 27–29).

A major hurdle for the improvement of the yield of **4a** was the competing reductive elimination pathway to β-lactam **5a**. To

## Table 1 | Optimization of the reaction conditions[a,b]

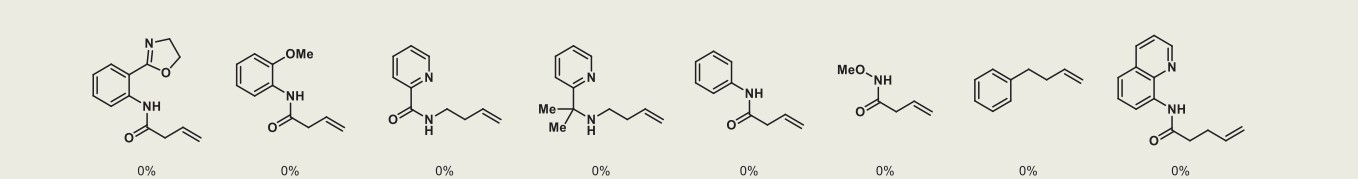

| Entry | Catalyst (10 mol%) | Solvent | Additive | Yield (%) of 3a+4a (r.r. for 3a/4a)[c] | Yield (%) of 5a | Yield (%) of recovered 1a |
|---|---|---|---|---|---|---|
| 1 | Cu(TFA)$_2$·H$_2$O | DCE | – | 0 | 42 | 12 |
| 2 | Cu(TFA)$_2$·H$_2$O | Toluene | – | 16 (1:3) | 25 | 21 |
| 3 | Cu(TFA)$_2$·H$_2$O | Dioxane | – | 0 | 73 | 0 |
| 4 | Cu(TFA)$_2$·H$_2$O | CH$_3$CN | – | 0 | 18 | 70 |
| 5 | Cu(TFA)$_2$·H$_2$O | NMP | – | 33 (10:1) | 10 | 20 |
| 6 | Cu(TFA)$_2$·H$_2$O | $^i$PrOH | – | 45 (>20:1) | 10 | 31 |
| 7 | Cu(TFA)$_2$·H$_2$O | $^i$BuOH | – | 65 (>20:1) | 0 | 20 |
| 8 | Cu(NO$_3$)$_2$ | $^i$BuOH | – | Trace | 0 | 90 |
| 9 | Cu(OAc)$_2$·H$_2$O | $^i$BuOH | – | 72 (>20:1) | 0 | 10 |
| 10 | CuI | $^i$BuOH | – | 47 (>20:1) | 0 | 0 |
| 11 | Cu(CH$_3$CN)$_4$PF$_6$ | $^i$BuOH | – | 56 (>20:1) | 0 | 0 |
| 12 | Cu(OAc)$_2$·H$_2$O | $^i$BuOH/DCE[d] | – | 76 (>20:1) | 0 | 5 |
| 13 | Cu(OAc)$_2$·H$_2$O | $^i$BuOH/dioxane[d] | – | 71 (>20:1) | 0 | 0 |
| 14 | Cu(OAc)$_2$·H$_2$O | $^i$BuOH/CH$_3$CN[d] | – | 82 (>20:1) | 0 | 5 |
| 15 | – | $^i$BuOH/CH$_3$CN[d] | – | 0 | 0 | 92 |
| 16 | Ni(cod)$_2$ | $^i$BuOH/CH$_3$CN[d] | – | 0 | 0 | 85 |
| 17 | Co(acac)$_2$ | $^i$BuOH/CH$_3$CN[d] | – | 0 | 0 | 83 |
| 18 | Cu(TFA)$_2$·H$_2$O | Xylene[e] | – | 25 (1:5) | 28 | 30 |
| 19 | Cu(TFA)$_2$·H$_2$O | Benzene[e] | – | 15 (1:8) | 67 | 5 |
| 20 | Cu(TFA)$_2$·H$_2$O | Benzene[e] | DBU (0.5 eq) | 0 | 0 | 90 (1a') |
| 21 | Cu(TFA)$_2$·H$_2$O | Benzene[e] | CS$_2$CO$_3$ (0.5 eq) | 0 | 0 | 62 + 25 (1a') |
| 22 | Cu(TFA)$_2$·H$_2$O | Benzene[e] | Mg(OAc)$_2$ (0.4 eq) | 34 (<1:20) | 55 | 0 |
| 23 | Cu(TFA)$_2$·H$_2$O | Benzene[e] | Zn(OAc)$_2$ (0.4 eq) | 45 (<1:20) | 32 | 8 |
| 24 | Cu(TFA)$_2$·H$_2$O | Benzene[e] | Mn(OAc)$_3$·3H$_2$O (0.4 eq) | 37 (<1:20) | 36 | 0 |
| 25 | Cu(TFA)$_2$·H$_2$O | Benzene[e] | Zn(OAc)$_2$ (1.0 eq) | 35 (<1:20) | 15 | 28 |
| 26 | Cu(TFA)$_2$·H$_2$O | Benzene[e] | ZnBr$_2$·TMEDA (0.4 eq) | 20 (1:10) | 60 | 0 |
| 27 | Cu(OAc)$_2$·H$_2$O | Benzene[e] | Zn(OAc)$_2$ (0.4 eq) | 37 (<1:20) | 38 | 0 |
| 28 | Cu(CH$_3$CN)$_4$PF$_6$ | Benzene[e] | Zn(OAc)$_2$ (0.4 eq) | 20 (<1:20) | 40 | 30 |
| 29 | Fe(OTf)$_2$ | Benzene[e] | Zn(OAc)$_2$ (0.4 eq) | 0 | 0 | 89 |
| 30 | Cu(TFA)$_2$·H$_2$O | Benzene[e] | Zn(OAc)$_2$ (0.4 eq) + TFA (1.0 eq) | 62 (<1:20) | 13 | 0 |
| 31 | Cu(TFA)$_2$·H$_2$O | Benzene[e] | Zn(OAc)$_2$ (0.4 eq) + TFA (2.0 eq) | 70 (<1:20) | 0 | 0 |
| 32 | Cu(TFA)$_2$·H$_2$O | Benzene[e] | Zn(OAc)$_2$ (0.4 eq) + HOAc (2.0 eq) | 52 (<1:20) | 20 | 8 |
| 33 | Cu(TFA)$_2$·H$_2$O | Benzene[e] | Zn(OAc)$_2$ (0.4 eq) + TfOH (2.0 eq) | 0 | 0 | 30 |

TFA trifluoroacetic acid, DCE dichloroethane, NMP N-methylpyrrolidone, DBU 1,8-diazabicyclo[5.4.0]undec-7-ene, cod 1,5-cyclooctadiene, acac acetylacetone, TMEDA N,N,N',N'-tetramethylethylenediamine.
[a]Conditions: **1a** (0.20 mmol), **2a** (0.50 mmol), and metal catalyst (0.02 mmol) dissolved in solvent (3.0 mL) at 90 °C for 3 h.
[b]Isolated yields.
[c]E/Z is >20:1 if not stated otherwise.
[d]2.5 mL $^i$BuOH + 0.5 mL other solvent.
[e]In 2.0 mL solvent for 4 h.

overcome this issue, a variety of external additives were tested. Eventually, it was unexpectedly found that the addition of TFA would produce **4a** in 62% yield, and the yield of β-lactam **5a** decreased to 13% (entry 30). Increasing the amount of TFA from 1.0 to 2.0 equiv further improved the yield of **4a** to 70%, and the reductive elimination process was completely supressed (entry 31, as conditions B). We have also tried to carry out the reaction with other Brønsted acids (entries 32–33). Whereas no better results were obtained, and when TfOH was added, the reaction system became messy with partial recovery of the alkene substrate (30%). It was noteworthy that treatment of alkenes bearing other bi- or monodentate auxiliaries rather than AQ amide[58–75], 4-phenylbutene, or γ–δ unsaturated AQ amide under the conditions A–C all failed to give either coupling or reductive elimination products. The presence of AQ as a bidentate auxiliary could not only simulate the intermolecular reaction as an intramolecular variant to enhance the reactivity of nonactivated olefins, but also stabilize the high-valent copper species via a [5,5]-metallabicyclic intermediate to allow for further hydride elimination or reductive elimination. Moreover, we wondered whether an addition of external ligand would compete with AQ to coordinate with the metal center, and thus influence the reaction. In this context, several ligands were tested. To our delight, the external ligands had little effect on the regio- and diastereoselectivities of the Heck-type reactions, but the reaction yields decreased in varying degrees (for details, see Supplementary Fig. 1).

## Substrate scope

With the optimized conditions in hand, the coupling reactions between various *N*-fluoro-sulfonamides and alkenyl amide **1a** were investigated. As shown in Fig. 3, the *N*-fluoro-sulfonamides bearing tertiary δ-C–H bonds were first tested under conditions A or B to deliver the coupling products with quaternary carbon centers. The substituents on the phenyl ring of arylsulfonamide group had little impact on the reactivity to produce vinyl products **3b**–**3f** under conditions A or allylic products **4b**–**4f** under conditions B. Treatment of *N*-fluoro-2-naphthalene- or methane-sulfonamides with conditions A or B afforded **3g**, **3h** or **4g**, **4h**, respectively, with excellent regio- and *E*-stereoselectivities. The *N*-fluoro-sulfonamides containing branched alkyl chain were smoothly transformed to the corresponding vinyl products **3i**–**3m** or allylic products **4i**–**4m**. Of note, the tertiary δ-C–H bond could be selectively functionalized in the presence of a secondary δ′-C–H bond, affording products **3l** or **4l** in yields of 80 and 68%, respectively. Moreover, the coupling reactions were compatible with electronically diverse phenyl groups (**3n**, **3o** or **4n**, **4o**) bearing either electron-donating (−OMe, **3p**, **3s** or **4p**, **4s**) or -withdrawing (−Br or −CF₃, **3q**, **3r** or **4q**, **4r**) substituents. It should be mentioned that chlorine atom (**3t** or **4t**), azide (**3u** or **4u**), ether (**3v** or **4v**), ester (**3w**, **3x** or **4w**, **4x**), and cyclopropane (**3y** or **4y**) were all well-tolerated without any sign of functional-group interconversion. Consistent with the aforementioned result, *N*-fluoro-sulfonamides underwent the coupling reactions preferentially at the tertiary δ-C–H bonds over the secondary ones (marked with an orange circle). Apart from δ,δ-dimethyl substituted fluorosulfonamides, the δ-methyl-δ-pentyl, δ-ethyl-δ-butyl, and δ,δ-cyclobutyl substituted *N*-fluoro-sulfonamides were found to couple with alkenyl amide **1a** to afford vinyl products **3z**–**3ab** or allylic products **4z**–**4ab**. Subjection of *N*-fluoro-sulfonamides derived from lithocholic acid or Celecoxib to reaction conditions A produced **3ac** and **3ad** in 60 and 78% yields, respectively, while **4ac** and **4ad** were obtained in the yields of 52 and 67%, respectively, under reaction conditions B; showing the synthetic potential of this methodology for late-stage modification of bioactive molecules.

Subsequently, a variety of *N*-fluoro-sulfonamides bearing secondary δ-C–H bond were investigated under conditions D or B (Fig. 4). Both linear fluorosulfonamides with different alkyl chain lengths and the branched fluorosulfonamides worked well, and vinyl products **3ae**–**3aj** were obtained as single regioisomers with moderate to good

stereoselectivity (*E/Z* ranging from 5:1 to >20:1), while allylic products **4ae**–**4aj** were isolated as *E*-isomers with excellent regioselectivity (*r.r.* ranging from 10:1 to >20:1). The secondary C–H bond could be activated prior to a primary one (marked with a purple circle) to give products **3ak** or **4ak** in yields of 65 and 62%, respectively. Benzylic δ-C–H bond proved to be available for the coupling reaction, giving rise to products **3al**–**3ap** or **4al**–**4ap**. With regard to the functional-group tolerance, all of the halogen atom, ester, and even terminal olefin could suffer from either conditions D or B to deliver vinyl products **3aq**–**3as** or allylic products **4aq**–**4as**, providing a useful handle for further manipulation. However, *N*-fluoro-sulfonamide bearing an azide group smoothly decomposed, and failed to give the corresponding allylic coupling product **4at** under conditions B, a result different from that of **4u** from tertiary δ-C–H bond. This may be attributed to the adverse effect of azide group on allylic coupling reactions (only 45% yield for **4u**) as well as the less stability of secondary alkyl radical intermediates. Cyclic fluorosulfonamides with 5-, 6-, or even 7-membered ring could participate in the coupling reaction to provide **3au**–**3aw** or **4au**–**4aw**, and no diastereoselectivity was observed (*d.r.* = 2:1 or 1:1). When *N*-fluoro-sulfonamide derived from 1-C-ethylaminoadamantane was employed, coupling products **3ax** or **4ax** were obtained in 77 and 68% yields, respectively. For complex biologically relevant fluorosulfonamide, such as that from (+)-dehydroabietylamine, the coupling reaction occurred regioselectively at the C10 position to give **3ay** or **4ay** with complete diastereocontrol (*d.r.* > 20:1). Subjecting the *N*-fluoro-sulfonamide containing both secondary δ-C–H bond and tertiary ε-C–H bond to the coupling reaction conditions D or B led to a mixture of δ-coupling products **3az** or **4az** (via 1,5-HAT) and ε-coupling products **3az′** or **4az′** (via 1,6-HAT)[76]. Unfortunately, neither vinyl nor allylic product was observed for the reaction of *N*-butyl fluorosulfonamide, presumably due to the less stability of primary carbon radical[77], which likely underwent decomposition prior to the addition to the olefin partner.

Encouraged by the aforementioned results, we then evaluated the scope of unactivated alkenes for the access to vinyl coupling products under slightly modified reaction conditions E, wherein isopropanol proved to be the best choice of solvent (Fig. 5). A variety of alkenyl amides bearing α-substituents were successfully converted into the corresponding vinyl products. α-Methyl or n-butyl substituted alkenes led to the formation of **6a** and **6b** in 84 and 88% yields, respectively, while bulky α-isopropyl substituent gave a diminished yield of 70% for **6c**, all being produced as single regio- and *E*-stereoisomer. The alkenes bearing electronically diverse phenyl groups with either electron-donating (−OMe) or -withdrawing (−Br, −Cl) substituents provided excellent yields of the target vinyl products **6d**–**6g**. The structure of **6d** was further confirmed by X-ray crystallographic analysis (CCDC 2163753 contains the supplementary crystallographic data for compound **6d**). Besides phenyl group, other aryl groups including naphthyl (**6h**, **6i**) and thiophene-yl (**6j**) were compatible with the reaction conditions E. Significantly, the reaction proceeded well in the presence of diverse functionalities, such as acetal (**6k**), halogen atom (**6l**), ether (**6m**), cyclopropyl (**6n**), and even free hydroxyl (**6o**). For diene or enyne substrates, the β–γ double bond was selectively activated to produce **6p**–**6s** in good yields, and the remaining γ–δ or δ–ε double bonds as well as the acetylene could serve as potential handles for further transformation. Subjection of the sterically hindered α,α-disubstituted alkenes to the reaction smoothly afforded **6t**–**6v** in excellent yields, wherein the steric effect made little influence on the reaction outcome. For 1,1-disubstituted alkenes, the vinyl coupling products were obtained along with the regioisomers generated from β−elimination of the hydrogen atom on R² alkyl chain, and the mixtures **6w** (*r.r.* = 2:1) and **6x** (*r.r.* = 2:3) were obtained in 82 and 65% yields, respectively. To our delight, the internal alkenes typically suffering from low substrate reactivity and poor product stereoselectivity[78] proved to be suitable partners; both cyclic and *Z*-

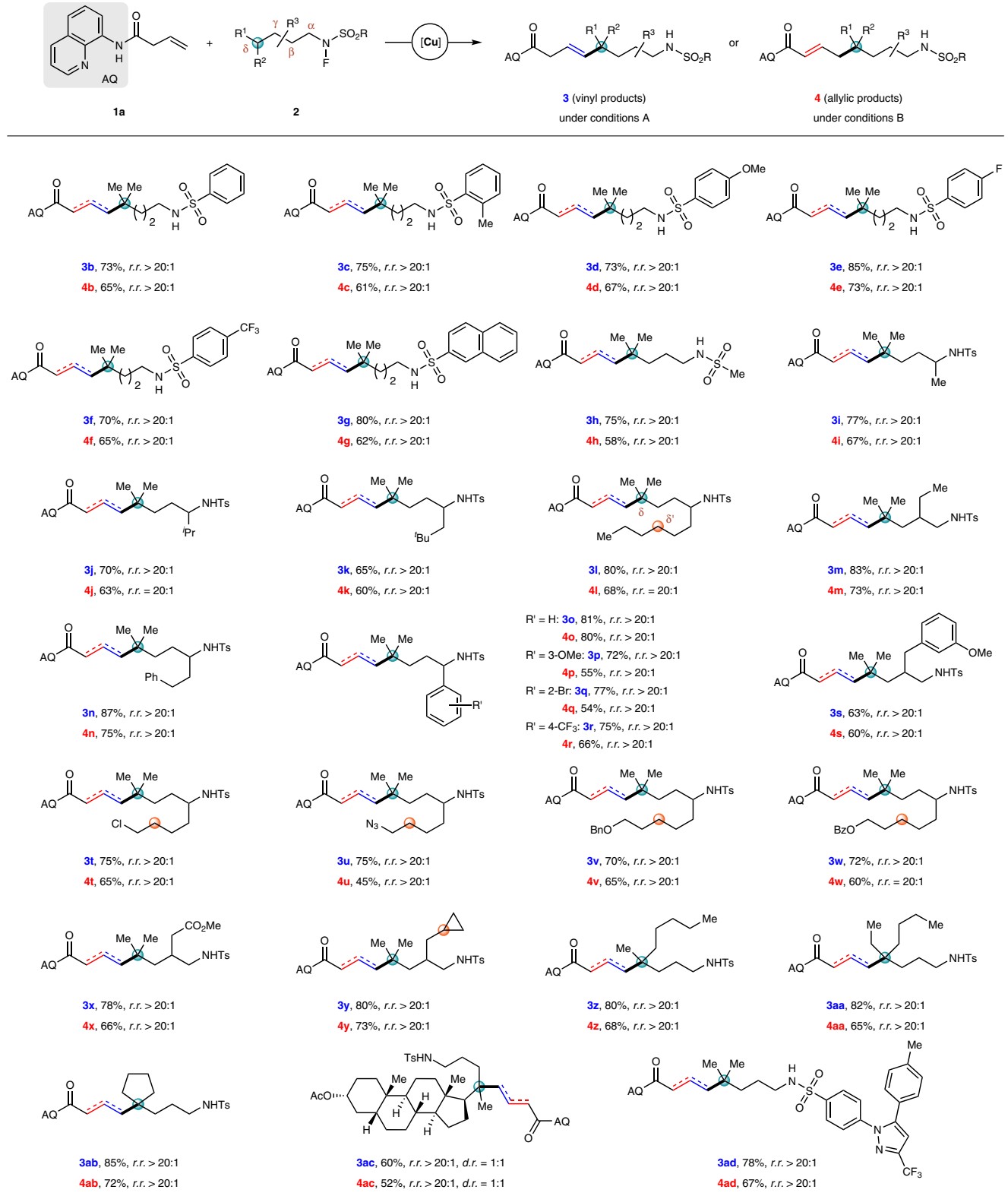

**Fig. 3 | Substrates scope of *N*-fluoro-amides bearing tertiary δ-C−H bonds.**
Conditions A (for vinyl coupling): **1a** (0.20 mmol), **2** (0.50 mmol), and Cu(OAc)₂·H₂O (0.02 mmol) in a mixed *ᵗ*BuOH/CH₃CN (2.5/0.5 mL) at 90 °C for 3 h. Conditions B (for allylic coupling): **1a** (0.20 mmol), **2** (0.50 mmol), Cu(TFA)₂·H₂O (0.02 mmol), Zn(OAc)₂ (0.08 mmol), and TFA (0.40 mmol) in benzene (2.0 mL) at 90 °C for 4 h. Isolated yields are shown. *E/Z* is >20:1 if not stated otherwise. TFA trifluoroacetic acid.

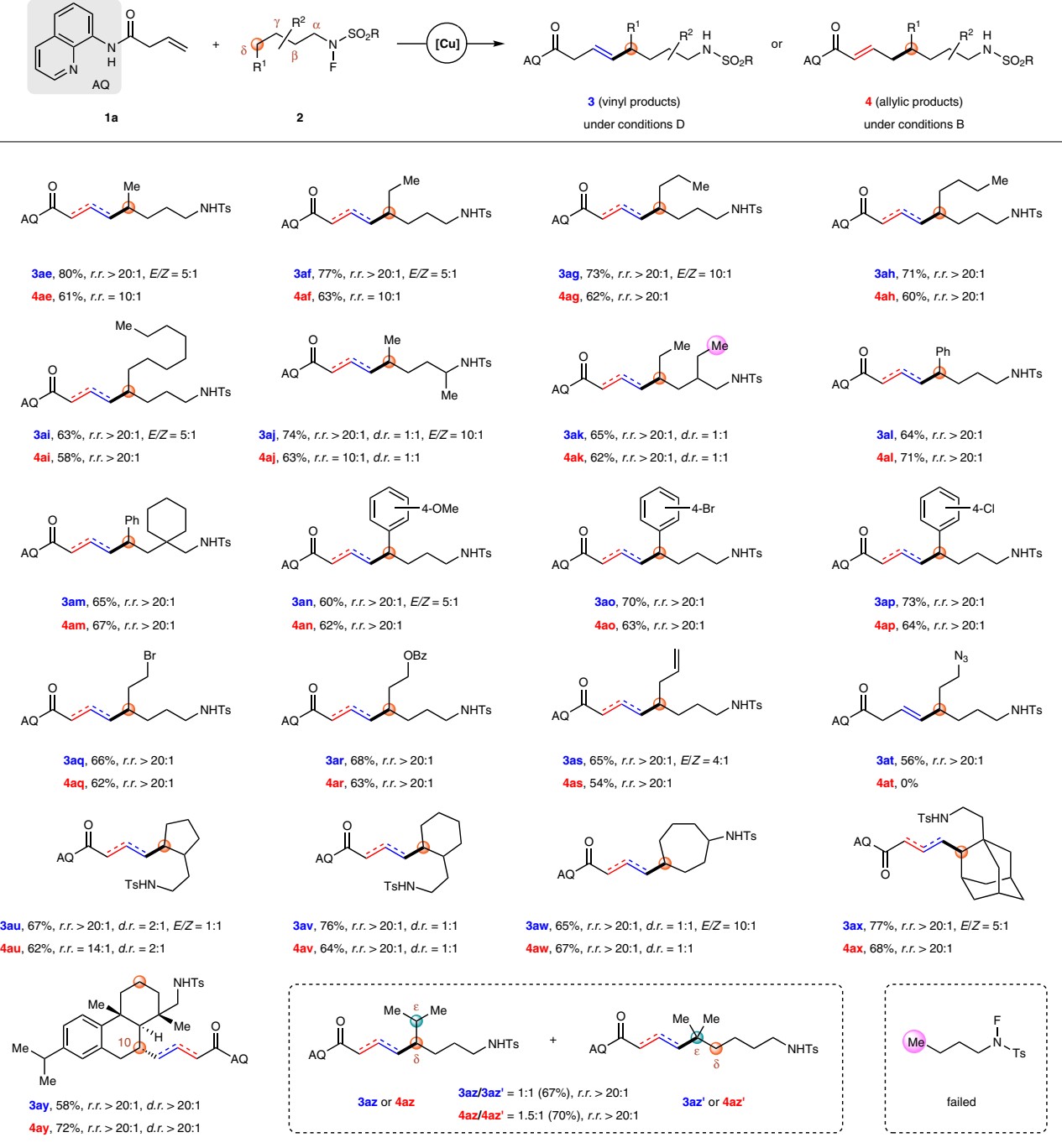

**Fig. 4 | Substrates scope of *N*-fluoro-amides bearing secondary δ-C–H bonds.** Conditions D (for vinyl coupling): **1a** (0.20 mmol), **2** (0.50 mmol), and Cu(OAc)₂·H₂O (0.02 mmol) in ᵗBuOH/dioxane (2.5/0.5 mL) at 90 °C for 3 h; conditions B (for allylic coupling): **1a** (0.20 mmol), **2** (0.50 mmol), Cu(TFA)₂·H₂O (0.02 mmol), Zn(OAc)₂ (0.08 mmol), and TFA (0.40 mmol) in benzene (2.0 mL) at 90 °C for 4 h. Isolated yields are shown. *E/Z* is >20:1 if not stated otherwise. TFA trifluoroacetic acid.

internal alkenes were transformed into the corresponding trisubstituted alkenes **6y**–**6aa** with good reaction efficiency and excellent regio- and *E*-stereoselectivities.

Moving forward, we also tested the applicability of unactivated alkenes for the allylic coupling reaction under reaction conditions B. As shown in Fig. 6, the *Z*-internal alkene substrates could smoothly react with *N*-fluoro-sulfonamide **2a** to produce the desired products **7a** and **7b** as single regio- and *E*-stereoisomer. For 1,1-disubstituted alkene, however, the reaction led to a slow decomposition of the alkene substrate, and the desired allylic coupling product **7c** was not

detected. It was noteworthy that when α-substituted alkene, for example, the α-methyl alkenyl amide was subjected to the reaction, product **7d** was not obtained as expected. Instead, β-lactam **8** was isolated in 32% yield as a single *trans*-isomer with part of the alkene material recovered (30%). This may be attributed to the steric effect of the α substituent, which disfavors a coplanar transition state of the high-valent copper complex in concerted β–Hₐ elimination process, and thus alternative reductive elimination becomes the major reaction pathway to provide β-lactam **8** (for a detailed explanation by density functional theory (DFT) calculations, see Supplementary Fig. 2).

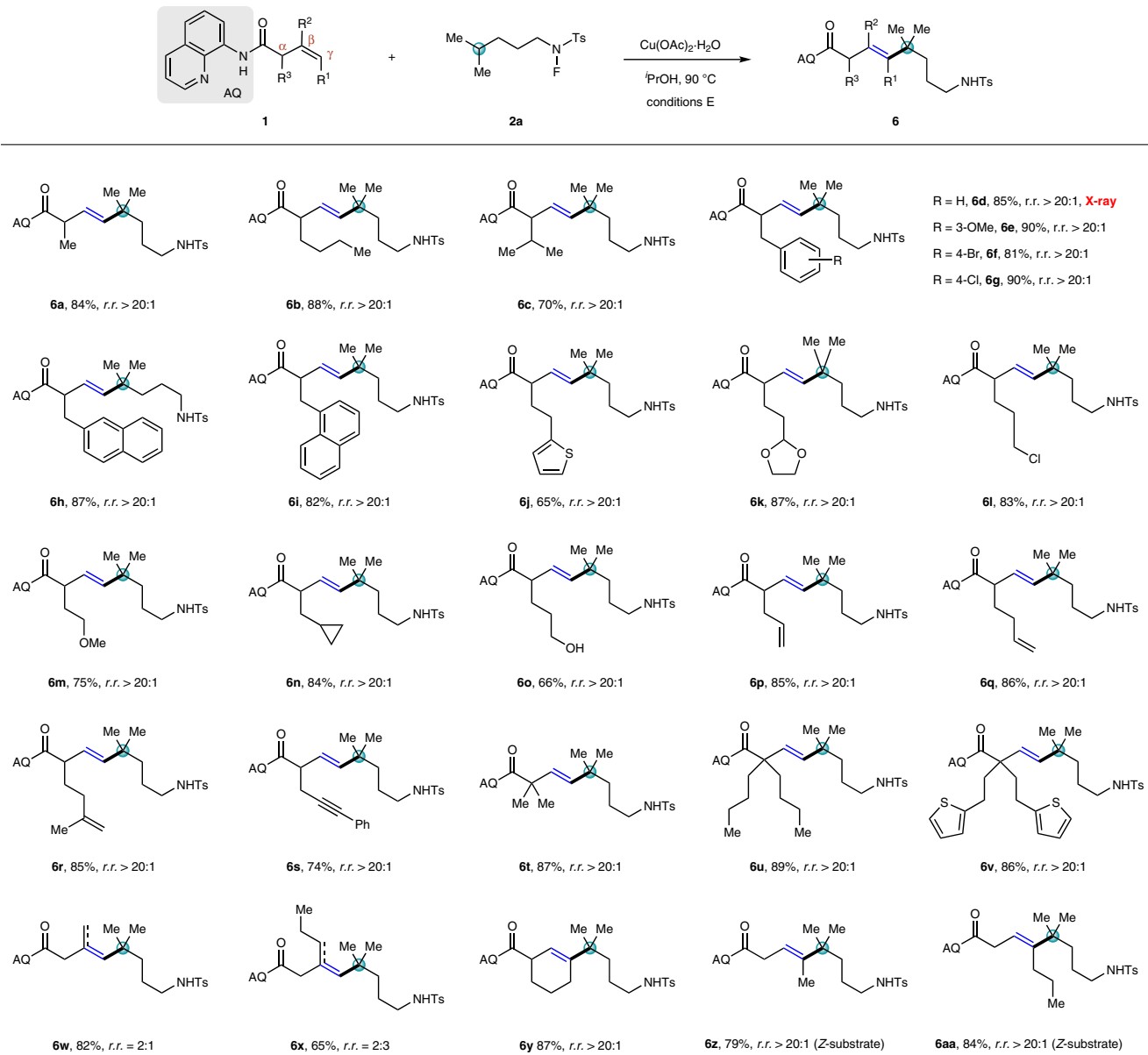

**Fig. 5 | Substrate scope of alkenes for vinyl coupling reactions.** Conditions E: **1** (0.20 mmol), **2a** (0.50 mmol), and Cu(OAc)$_2$·H$_2$O (0.02 mmol) in $^i$PrOH (3.0 mL) at 90 °C for 2 h. Isolated yields are shown. *E/Z* is >20:1 if not stated otherwise.

Of note, under reaction conditions C excluding the aforementioned promoting factors, the β-lactams **5** would dominate the product distribution through a reductive elimination of the high-valent copper intermediate[79], a process previously observed independently by Zhao[80,81], He/Chen/Wang[82,83], and Quan/Liang/Wang[43] in alkylamination of unactivated alkenes. As demonstrated in Fig. 6, we briefly investigated the scope for the synthesis of β-lactams. Both the α-substituted and internal alkenes could react with *N*-fluoro-sulfonamide **2a**, and the corresponding β-lactams **5b**–**5e** were obtained in yields ranging from 63 to 85% with excellent diastereoselectivity (*d.r.* > 20:1). Subsequently, alkenyl amide **1a** was utilized as the alkene component and tested in combination with different *N*-fluoro-sulfonamides. Both linear and branched *N*-fluoro-sulfonamides bearing tertiary or secondary δ-C−H bonds were successfully converted into the corresponding β-lactams **5f**–**5j** in good yields. This reaction showed good compatibility with the fluorosulfonamides derived from complex bioactive molecules. For example, treatment of the *N*-fluoro-sulfonamide prepared from

(+)-dehydroabietylamine with alkenyl amide **1a** under reaction conditions C provided the desired β-lactam **5k** in 75% yield.

### Control experiments

To better understand the reaction mechanism, several control experiments were performed (Fig. 7). The crossover experiments using *N*-fluoro-sulfonamides (**2a** or **2al**) and the corresponding simple sulfonamides (**9** or **10**) were conducted under conditions A and D, respectively. No crossover products were observed (Fig. 7a). By employing NFSI as an external amidyl radical precursor[84], we examined intermolecular coupling reactions between alkenyl amide **1a** and simple sulfonamides **9** or **10** under conditions A and D, respectively, but neither vinyl coupling products **3a** nor **3al** were detected (Fig. 7b). These results could rule out the involvement of an intermolecular hydrogen atom abstraction process in the reaction. To further conform an intramolecular hydrogen migration process, deuterium-labeling experiments using external CD$_3$OD were carried out with **1a**

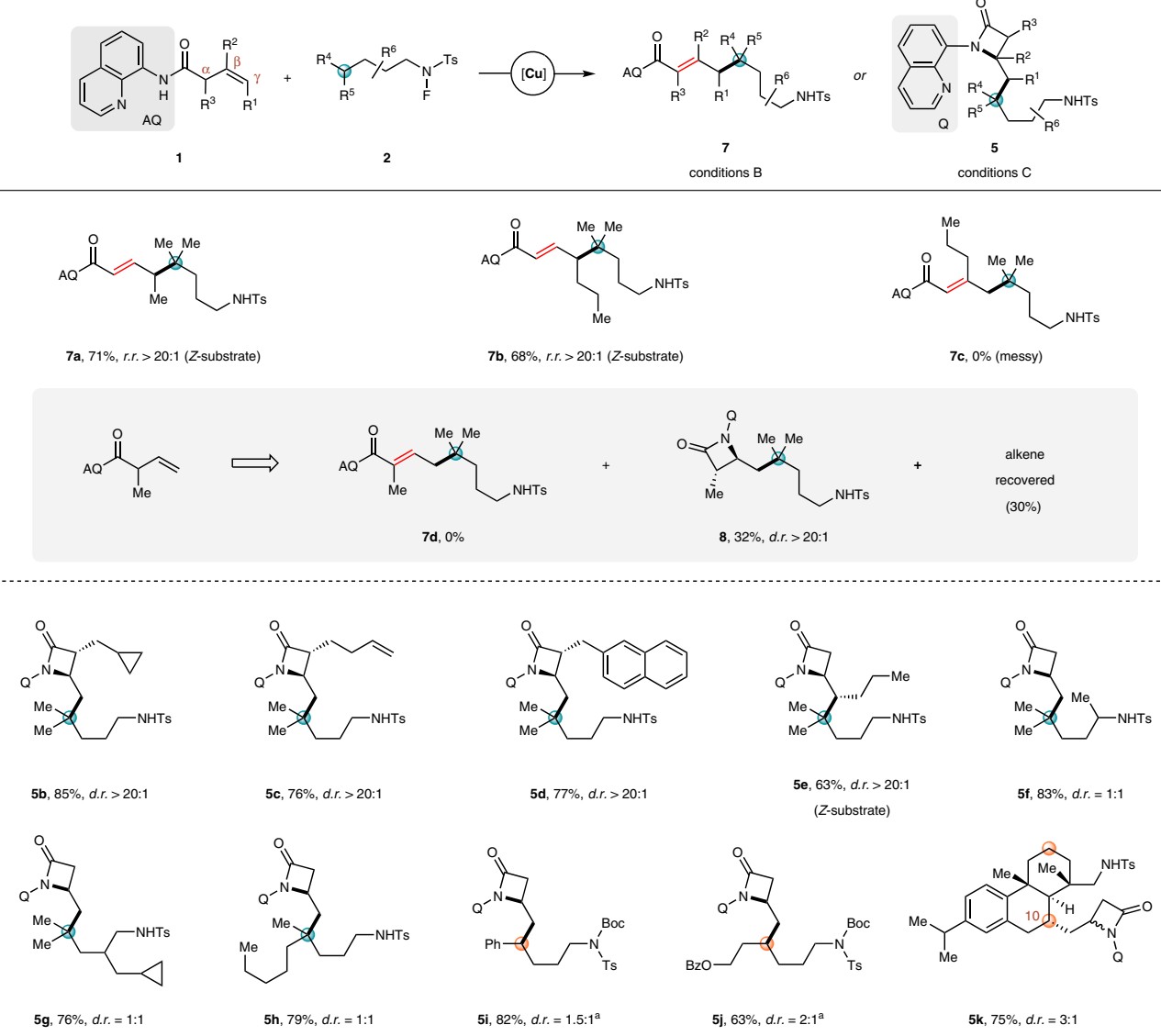

**Fig. 6 | Substrate scope of alkenes for allylic coupling reactions and for the synthesis of β-lactams.** Conditions B: **1** (0.20 mmol), **2a** (0.50 mmol), Cu(T-FA)$_2$·H$_2$O (0.02 mmol), Zn(OAc)$_2$ (0.08 mmol), and TFA (0.40 mmol) in benzene (2.0 mL) at 90 °C for 4 h. Conditions C: **1** (0.20 mmol), **2a** (0.50 mmol), and Cu(TFA)$_2$·H$_2$O (0.02 mmol) in dioxane (3.0 mL) at 90 °C for 3 h. Isolated yields are shown. E/Z is >20:1 if not stated otherwise. TFA trifluoroacetic acid. [a]The yield was recorded after a Boc-protection of the crude secondary amine product.

and **2a** under either conditions A′ or B′. As expected, no deuterium was incorporated into the vinyl or allylic coupling products (Fig. 7c).

When 3.0 equiv of either 2,2,6,6-tetramethylpiperidinooxy or butylated hydroxytoluene was added as a radical scavenger into the reaction of **1a** and **2a**, the desired coupling reactions were completely suppressed under either conditions A′′ or B′′. For radical clock experiments[85], subjection of N-fluoro-sulfonamide **11** bearing a ε-ζ terminal double bond to the reaction with alkenyl amide **1a** under conditions B generated diene **12**, and the double bond migrated from ε-ζ to δ-ε position potentially via an π-allylic radical intermediate. Treatment of N-fluoro-sulfonamide **13** tethering a remote terminal double bond with **1a** under conditions B provided allylic coupling product **14** in 63% yield via an intramolecular radical cyclization followed by an intermolecular Heck-type reaction. Moreover, when alkene substrate **15** containing a β-cyclopropyl moiety reacted with N-fluoro-sulfonamide **2a** under conditions E, only trace amount of the desired coupling product **16** was detected, and instead acetate **17** and seven-membered lactam **18** were produced in 9 and 62% yields, respectively, via ring-opening of the cyclopropyl group. Taken

together, these results demonstrated that the reaction involved alkyl radical intermediates (Fig. 7d).

Subjecting vinyl product **3a** to either conditions A or B in the absence of N-fluoro-sulfonamide did not provide any allylic product **4a**, only with the vast majority of the material recovered, which could exclude the possibility that the allylic products were generated from the corresponding vinyl products via olefin isomerization (Fig. 7e). Moreover, the present strategy could be further applied to other types of alkyl electrophiles. For example, the reaction of our previously employed cycloketone oxime ester[41,83,86] with alkene **1a** under conditions B delivered the desired allylic product **19** in 72% yield as a single regio- and E-stereoisomer, while β-lactam **20** was obtained in 68% yield under conditions C (Fig. 7f). Considering the distinct models to generate alkyl radicals from N-fluoro-sulfonamide and cycloketone oxime ester, these results have demonstrated that after initiating the SET process of radical precursors, the subsequent generation of alkyl radicals had limited effect on the following formation of Cu(III) intermediate and hydride elimination or reductive elimination.

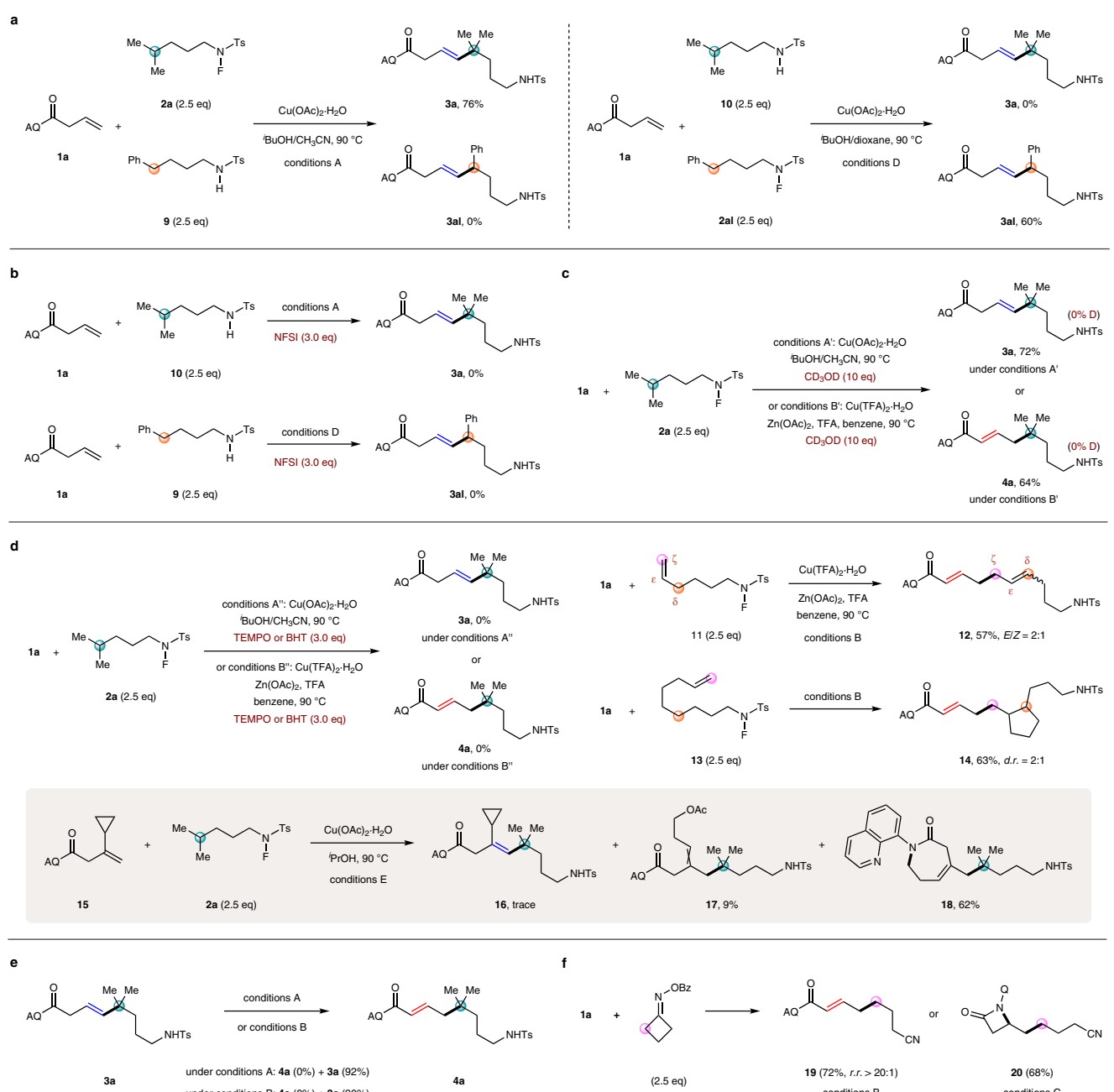

**Fig. 7 | Control experiments. a** Crossover experiments. **b** With external amidyl radical precursor. **c** Deuterium-labeling experiments. **d** Radical trapping and radical clock experiments. **e** Evidence for no isomerization. **f** Extending to other alkyl electrophiles. TFA trifluoroacetic acid, NFSI *N*-fluorobenzenesulfonimide, TEMPO 2,2,6,6-tetramethyl-1-piperidinyloxy, BHT butylated hydroxytoluene.

## Proposed reaction mechanism

According to the above control experiment results and precedent literature, a plausible mechanism for this intermolecular Heck-type reaction was proposed, as shown in Fig. 8. Firstly, the disproportionation reaction of Cu(II) catalyst generated a Cu(I) species[87]. A ligand exchange between Cu(I) catalyst and alkenyl amide **1a** may occur to form complex **I** and release HX[34]. The coordination of C=C bond to Cu(I) center would reduce π electron density, and thus activate the olefin moiety. Subsequent SET of *N*-fluoro-sulfonamide **2a** with copper complex **I** affords amidyl radical **II** and Cu(II) intermediate **III**[88], and the released fluorine anion is proposed to be captured by the protons due to the strong interaction of H–F bond[54,89,90]. The resultant amidyl radical **II** is a high-energy species[77,91,92], which undergoes an intramolecular HAT with

remote δ-C(sp³)–H bond to form a carbon-centered radical **IV**[93]. Radical addition of **IV** to the olefin moiety of Cu(II) intermediate **III** results in the formation of a secondary carbon-centered radical, which can rapidly recombine with the nearby Cu(II) center to produce the key Cu(III) intermediate **V**[34,41,43,45,46,49,50,53,80–83]. At this point, there are three potential reaction pathways from **V** to provide different products depending on the choice of external additives. These three paths include a concerted β–H_b elimination (path 1) to form vinyl product **3a** in alcohol solvent, a concerted β–H_a elimination (path 2) to afford allylic product **4a** under acidic conditions in the presence of carboxylate, and a C–N reductive elimination (path 3) to generate β-lactam **5a** without the aforementioned promoting factors. Along with the formation of the coupling or reductive elimination products, the Cu(I) catalyst can be

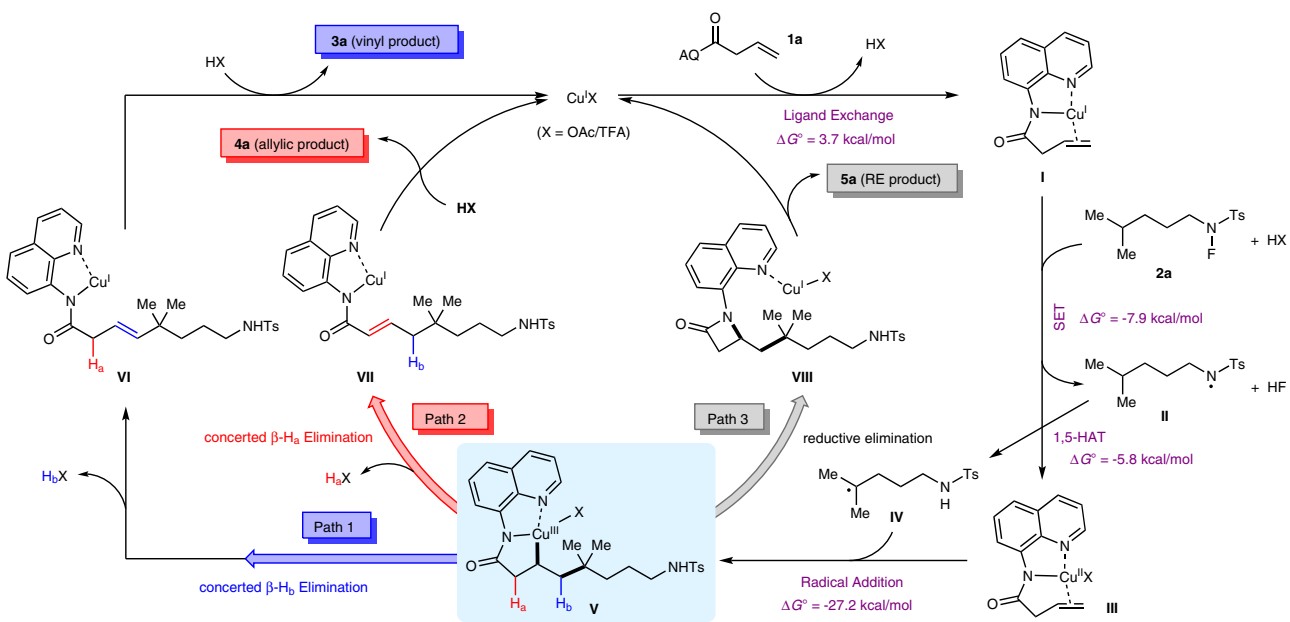

**Fig. 8 | The proposed mechanism.** Three plausible different reaction pathways for the intermolecular Heck-type reaction of unactivated alkenes and *N*-fluoro-sulfonamides.

regenerated to restart the catalytic cycle. Here, the origin of the complete switch of regioselectivity by the choice of different additives is worth investigating.

## Theoretical calculations

To gain a deeper understanding of the divergent regioselectivities from Cu(III) species **V**, especially the key role of promoting factors, three possible reaction mechanisms with different additives have been evaluated through DFT calculations at the SMD[94]($^i$BuOH/benzene/dioxane)/(U)M06[95]/[6-311++G(d,p)/SDD[96](Cu)]//SMD($^i$BuOH/benzene/dioxane)/(U)M06/[6-31G(d)/LanL2DZ[97](Cu)] level (see the Supplementary Information for computational details). To ensure the rationality of the reaction mechanism prior to the regioselectivity-determining step from **V**, some elementary steps including ligand exchange, SET, 1,5-HAT, and radical addition were evaluated in alcohol solvent[98]. The DFT calculation results suggest that these four steps are accessible with the Gibbs free energy change ($\Delta G°$) values of 3.7, −7.9, −5.8, and −27.2 kcal/mol, respectively (Fig. 8).

Figure 9 and Supplementary Fig. 3 show the Gibbs energy profiles of three regioselectivity-determining steps (concerted β−$H_a$ elimination, concerted β−$H_b$ elimination and C−N reductive elimination) under different reaction conditions. In the conditions A, $^i$BuOH was employed as the reaction solvent based on entry 14 in Table 1. Initially, a Cu(III) complex **1'** can be formed through the ion exchange between counterion X of **V** and alkoxy group of $^i$BuOH. As the most favorable path, the alkoxy-assisted concerted β−$H_b$ elimination occurs through a five-membered-ring transition state **TS1** to afford Cu(I) intermediate **2'** with the small Gibbs activation energy ($\Delta G°^‡$) and the large negative $\Delta G°$ values of 10.0 and −17.2 kcal/mol, respectively (Fig. 9a). The concerted β−$H_b$ elimination can eventually lead to the formation of vinyl product **3a**. In addition, we also located the alternative concerted β−$H_a$ elimination and C−N reductive elimination, which lead to the allylic product **4a** and the β-lactam product **5a**, respectively. Although the participation of $^i$BuOH can also promote these two reaction steps with the $\Delta G°^‡$ values of 24.9 and 21.0 kcal/mol, respectively, they are still much more unfavorable than the above-concerted β−$H_b$ elimination.

In the conditions B, nonpolar benzene solvent was employed together with the presence of Zn(OAc)$_2$ and TFA based on entry 29 in Table 1. A stable Cu(III) complex **5'** can be generated through the Lewis

acid-base interaction between zinc and carbonyl (Fig. 9b). With the assistance of Zn(OAc)$_2$, concerted β−$H_b$ elimination (**5'→TS4**) and C−N reductive elimination (**5'→TS6**) from **5'** still require large $\Delta G°^‡$ values of 32.0 and 27.0 kcal/mol, respectively. However, the concerted β−$H_a$ elimination easily occurs through an eight-membered-ring transition state **TS5** to form an ionic Cu(I) complex **7'**, which is eventually transformed into the allylic product **4a**. The $\Delta G°^‡$ and $\Delta G°$ values of concerted β−$H_a$ elimination are 18.1 and 2.2 kcal/mol, respectively. The calculation results suggest that Zn(OAc)$_2$ additive promotes effectively the concerted β−$H_a$ elimination. On one hand, as a Lewis acid, Zn(OAc)$_2$ can reduce electron density of the electron-rich carbonyl group, improving the proton acidity of $H_a$. On the other hand, carboxylate anion can capture the protic $H_a$ via a conformationally strained metallacyclic transition state. If omitting the assistance of Zn(OAc)$_2$ in activating $H_a$, the energy barrier of concerted β−$H_b$ elimination becomes larger by 12.1 kcal/mol (**5'→TS7**). The addition of TFA may accelerate the turnover of catalytic amount of Zn(OAc)$_2$ via protonative demetalation, and thus suppress the competing reductive elimination process. It should be mentioned that when one $H_a$ is replaced by an alkyl group, for example, a methyl, the concerted β−$H_a$ elimination process switches to C−N reductive elimination (see Fig. 6). This is ascribed to the benefit of electron-donating alkyl group to reductive elimination, as well as the steric hindrance by α-methyl substitution, which is disadvantageous to a coplanar concerted β−$H_a$ elimination process (for details, see Supplementary Fig. 2).

In the conditions C, dioxane was used as the reaction solvent, and the promoting factors in conditions A and B were excluded based on entry 3 in Table 1. Different from the above two conditions, the direct C−N reductive elimination becomes the most favorable reaction pathway in the absence of any external additives (for details, see Supplementary Fig. 3). Specifically, the C−N reductive elimination of **9'** occurs through a three-membered-ring transition state **TS10** to form the three-coordinate Cu(I) intermediate **12'** with a $\Delta G°^‡$ value of 13.3 kcal/mol. Finally, the ligand exchange process can release the RE product **5a** and regenerate the Cu(I) catalyst.

Overall, the complete switch of regioselectivity by the choice of different additives has been reproduced by the potential energy surface calculations. When alcohol solvent is employed, the alkoxy group can coordinate with a copper center, and act as a Brønsted base to

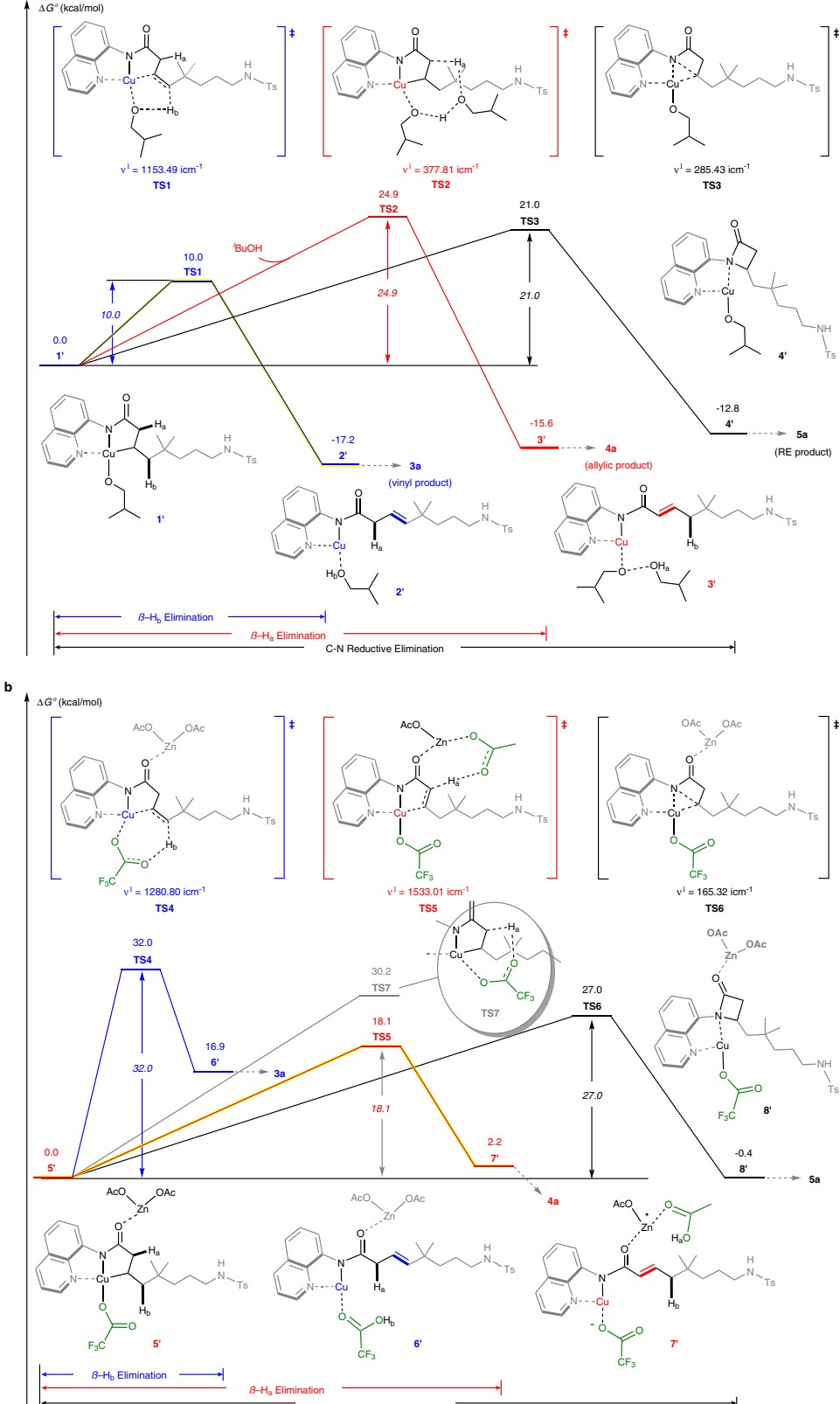

**Fig. 9 | Gibbs energy profiles (ΔG°) of three possible reaction pathways calculated at the SMD($^i$BuOH/benzene)/(U)M06/[6-311++G(d,p)/SDD(Cu)]//SMD($^{i-}$BuOH/benzene)/(U)M06/[6-31G(d)/ LanL2DZ(Cu)] level. a** Conditions A: formation of the vinyl product **3a** in $^i$BuOH solvent. **b** Conditions B: formation of the allylic product **4a** in the presence of TFA and Zn(OAc)$_2$.

facilitate the concerted β−H$_b$ elimination, leading to an access to the vinyl coupling product. In contrast, when metal carboxylate is added to a nonpolar solvent, the metal can serve as a Lewis acid to reduce electron density of carbonyl moiety to increase the proton acidity of β−H$_a$, then achieves concerted β−H$_a$ elimination through a carboxylate-participated eight-membered metallacyclic transition state to afford the allylic product. Moreover, the β-lactams dominate the product distribution through a favorable C−N reductive elimination of the Cu(III) intermediate in the absence of any aforementioned promoting factors.

## Discussion

In summary, a divergent regioselective intermolecular Heck-type reaction was reported via a directed copper-catalyzed coupling of *N*-fluoro-sulfonamides and unactivated alkenes. The complete regioselectivity switch has been realized by simply varying the external additives. This protocol featured with broad substrate scope for both terminal and internal alkenes, excellent regio- and stereoselectivities, good functional-group tolerance, and a simple operational procedure. The *N*-fluoro-sulfonamides proceeded through SET with copper catalyst followed by an intramolecular HAT to generate the corresponding carbon-centered radical intermediates, which then participated in the intermolecular coupling with alkenes. Further control experiments and DFT calculations demonstrated the involvement of alkyl radical intermediates and illustrated a detailed reaction profile of the key Cu(III) intermediate, highlighting the distinct effect of the external additives. In alcohol solvent, a hydroxyl-assisted concerted β−H$_b$ elimination led to the formation of vinyl coupling products, while the addition of metal carboxylate under acidic conditions resulted in a complete switch to concerted β−H$_a$ elimination through a carboxylate-involved eight-membered metallacyclic transition state to deliver the allylic coupling products; without these promoting factors, C−N reductive elimination of the Cu(III) species dominated the product distribution to form β-lactams. Furthermore, the present strategy could be extended to other alkyl electrophiles, such as cycloketone oxime esters. This report reveals the major factors contributing to distinguishing the detachable β-hydrogens, offering a reference model for following research in Heck-type reaction.

## Methods

### Synthesis of vinyl product 3a

To a dry Schlenk flask were added **1a** (42.4 mg, 0.20 mmol, 1.0 equiv), **2a** (137 mg, 0.50 mmol, 2.5 equiv), Cu(OAc)$_2$·H$_2$O (4.0 mg, 0.02 mmol, 0.10 equiv), anhydrous *i*BuOH (2.5 mL), and anhydrous CH$_3$CN (0.50 mL). The mixture was degassed for three times with argon and then stirred at 90 °C (oil bath) for 3 h. Once completion, the reaction was cooled to room temperature. The reaction mixture was filtered by celite, and the filtrate was concentrated in vacuo. Further purification by a flash column chromatography using eluents (PE/EA = 5:1) afforded the desired vinyl coupling product **3a** as yellow oil (76.3 mg, 0.16 mmol, 82%, *r.r.* > 20:1, *E/Z* > 20:1).

### Synthesis of allylic product 4a

To a dry Schlenk flask were added **1a** (42.4 mg, 0.20 mmol, 1.0 equiv), **2a** (137 mg, 0.50 mmol, 2.5 equiv), Cu(TFA)$_2$·H$_2$O (3.9 mg, 0.02 mmol, 0.10 equiv), Zn(OAc)$_2$ (14.7 mg, 0.08 mmol, 0.40 equiv), TFA (30 uL, 0.40 mmol, 2.0 equiv), and anhydrous benzene (2.0 mL). The mixture was degassed for three times with argon and then stirred at 90 °C (oil bath) for 4 h. Once completion, the reaction was cooled to room temperature. The reaction mixture was filtered by celite, and the filtrate was concentrated in vacuo. Further purification by a flash column chromatography using eluents (PE/EA = 5:1) afforded the desired allylic product **4a** as yellow oil (65.1 mg, 0.14 mmol, 70%, *r.r.* > 20:1, *E/Z* > 20:1).

### Synthesis of β-lactam 5a

To a dry Schlenk flask were added **1a** (42.4 mg, 0.20 mmol, 1.0 equiv), **2a** (137 mg, 0.50 mmol, 2.5 equiv), Cu(TFA)$_2$·H$_2$O (3.9 mg, 0.02 mmol, 0.10 equiv), and anhydrous dioxane (3.0 mL). The mixture was degassed for three times with argon and then stirred at 90 °C (oil bath) for 3 h. Once completion, the reaction was cooled to room temperature. The reaction mixture was filtered by celite, and the filtrate was concentrated in vacuo. Further purification by a flash column chromatography using eluents (PE/EA = 5:1) afforded the desired β-lactam **5a** as yellow oil (67.9 mg, 0.15 mmol, 73%).

## Data availability

The authors declare that all data supporting the findings of this research are available within the article and its Supplementary Information. Cartesian coordinates of the calculated structures are available in Supplementary Data 1. Crystallographic data can be obtained free of charge for **6d** (CCDC 2163753) from the Cambridge Crystallographic Data Centre (https://www.ccdc.cam.ac.uk/). Any further relevant data are available from the authors on request.

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

## Acknowledgements

We gratefully acknowledge the National Natural Science Foundation of China (21971034, for J.F.; 21702027, for J.F.; 22173016, for W.G.) and the Fundamental Research Funds for the Central Universities (2412019FZ017, for J.F.) for financial support.

## Author contributions

C.Z. and Y.L. designed and performed the experiments. Y.D. and W.G. performed the density functional theory calculations. M.L., D.X., and S.G. assisted in completing the experiments. Q.L. and Q.Z. participated in the data discussion. J.F. directed the project and wrote the manuscript. All the authors were involved in the interpretation of the results presented in the manuscript.

## Competing interests

The authors declare no competing interests.
