## [Peer Review File · Nature Communications]

REVIEWER COMMENTS

Reviewer #1 (Remarks to the Author):

In recent years, transition metal mediated radical Heck-type reactions of alkenes have been established as a powerful tool for functionalization of alkenes. Many catalytic variants have also been achieved to construct diverse C-C and C-N bonds by stereoselective intercept of the in situ-formed radical intermediates. However, the Heck-type reaction of unactivated alkenes started by alkyl radical derived from intermolecular 1,5-HAT is still unknown in spite of some examples involving direct coupling of alkyl radical. As a continuation of their recent success on using N-fluoro-sulfonamides as N-radical precursor, in this manuscript, Fu, Guan and collaborators report a case of copper-catalyzed intermolecular Heck-type reaction of unactivated alkenes and N-fluoro-sulfonamides with divergent regioselectivities controlled by additive effect with broad substrate scope and good to excellent regioselectivity. With fine tuning the reaction parameters, β -lactams via a C-N reductive elimination could be readily accessed. Alkylated N-fluoro-sulfonamides enjoyed the smooth 1,5-HAT, addition to alkenes and regioselective hydride elimination under the optimized conditions. Successful application of this protocol to a wide variety of coupling partners bearing various functional groups also highlights its synthetic potential. Remarkably, the scope of radical precursors can also be extended to other readily available starting materials. On the basis of combined experimental and computational studies, a reasonable mechanism involving the critical roles of 8-aminoquinoline assisted with diverse chelating additive is also proposed for the reaction. This work provides a useful strategy for further development of other types of radical-based Heck reaction of inactivated alkenes, and will surely be of great interest to the synthetic community.

Overall, the current study is well executed, the supporting information is complete, and the results are clearly presented. This work represents a significant advance in the field of copper mediated radical coupling. In conclusion, I believe the present work fully meet the standard of this journal. Thus, I support the publication of this work in Nature Communications after addressing some minor points.

- 1) This reaction showed quite a broad substrate scope with respect to the N-fluoro-sulfonamides and external and internal alkenes. Successful combination of simple alkyl group and inactivated alkene is very impressive. 8-aminoquinoline play an important role for the reaction to occur. Further explanation on this would improve the quality of this work.
- 2) In the reaction with cycloketone oxime ester as the substrate, equal level of selectivity and efficiency was observed, clearly indicating that this reaction should be compatible with more radical precursors, and I hope the author can explain this further.

3) Curious to know what would happen if privileged chiral ligands are used, such as BOX, PYBOX, BINAP etc?

4) This work represents an interesting advance in the chemistry of N-radical 1,5 HAT chemistry. A quite recent review paper about this chemistry might be included. E.g., *Chem. Rev.* 2022, 122, 2353. A related work on 1,n-HAT and Heck sequence based on Pd-catalysis might also be included. *Angew. Chem. Int. Ed.* 2019, 58, 1794–1798.

Reviewer #2 (Remarks to the Author):

In this work, Fu, Guan and co-workers reported a copper-catalyzed divergent regioselective Heck-type reaction of unactivated alkenes and N-fluoro-sulfonamides with a broad substrate scope (>100 examples) and good functional group tolerance. The complete control of regioselectivities was achieved by simply varying external additives, providing either vinyl products in alcohol solvent or allylic products in the presence of carboxylate and TFA. In addition, cyclization products dominated the reaction when excluding the promoting factors. Considering the importance of the Nobel-awarded Heck reaction in organic synthesis, systematic studies on the remaining challenges of this chemistry is awfully fascinating. The present work focuses on the control of regioselectivity in Heck-type reaction of unactivated alkenes, one recognized challenge as discussed in Figure 1. It is fairly impressive for a precise distinction of the several detachable β -hydrogens, and the authors have provided a satisfactory explanation based on a [5,5]-metallabicyclic intermediate through DFT calculations.

Based on the aforementioned advantages, I recommend this work to be published on Nature Communications after minor revision, and some comments are shown below:

1)The authors stated that this is the first time for N-fluoro-amides to be applied in the coupling with alkenes. Aside from N-fluoro-sulfonamides, can other types of alkyl electrophiles be used in this divergent regioselective Heck-type reaction?

2)For 4at, the allylic coupling of secondary alkyl radical intermediate and unactivated alkene failed. However, the reaction of tertiary alkyl radical intermediate bearing an azide group can lead to 4u. It is better to give a brief explanation for this special case.

3)The β -lactam is a unique scaffold considering its bioactivity and presence in biologically relevant molecules. A representative review (*Org. Biomol. Chem.* 2018, 16, 6840-6852) should be cited to highlight the synthesis of β -lactams.

4)A recent DFT computational work investigated the mechanism of C–O bond activation of butanoic acid substrates bearing the 8-aminoquinoline group catalyzed by Pd(OAc)₂ (*Inorg. Chem.* 2021, 60, 17555-17564). Although the employed metal catalyst and substrate are different from this work,

such theoretical insights into the activation of strong alkyl C(sp³) covalent bonds via the 8-aminoquinoline directing strategy should be cited.

5) Imaginary frequencies of the transition states on the PES should be provided in Figure 6.

Reviewer #3 (Remarks to the Author):

In this manuscript, Fu, Guan, and coworkers developed the divergent regioselective intermolecular Heck-type reaction via a directed copper-catalyzed coupling of N-fluoro-sulfonamides and unactivated alkenes. Recently, they have reported the intermolecular Heck-type reactions of unactivated alkenes with alkyl electrophiles, including α -bromo esters in 2018 (ref. 34) and cycloketone oxime esters in 2020 (ref. 41). In this work, they extended this chemistry to N-fluoro-sulfonamides, and successfully realized divergent regioselective Heck-type reaction. As we know, it's difficult to adjust the divergent regioselectivities from a highly reactive Cu(III) intermediate, and the β -lactams are commonly obtained (ref. 43-47). The Heck-type vinyl coupling products are only observed in special cases.

Herein, the authors realized the complete regioselectivity by simply varying the external additives. Control experiments and DFT calculations illustrated a detailed reaction profile of the key Cu(III) intermediate, highlighting the distinct effect of the external additives. Moreover, this protocol shows a broad substrate scope for both alkenes and structurally diverse N-fluoro-sulfonamides, producing the corresponding products with excellent regio- and stereoselectivities. More than 150 examples are provided, showing the broad scope, as well as some limitations of the substrates. And the manuscript and SI is well organized. Overall, this reviewer supports the acceptance of the manuscript after a minor revision.

1. Have the authors tried the γ - δ unsaturated amide substrates, for the three kinds of reaction pathways?
 2. Does the authors have tried some other types of transition metals, such as the earth-abundant nickel or cobalt salt?
 3. For the control experiment in Figure 3F, C-N reductive elimination reaction by using 1a and cycloketone oxime ester as substrates was suggested to be added.
 4. In the proposed mechanisms (Figure 4), the reaction was initiated by Cu(I) species, rather the added Cu(II) catalyst. Please explain why. is there any rationale or reference?
 5. The large-scale experiment should be added in the manuscript and SI.
 6. Line 117: change (r.r. < 20:1) to (r.r. > 20:1);
- In SI, Page S286: the signal-to-noise for ¹³C NMR of compound 3at is poor. Please check it.

Response to the Reviewer 1's comments

Comments:

In recent years, transition metal mediated radical Heck-type reactions of alkenes have been established as a powerful tool for functionalization of alkenes. Many catalytic variants have also been achieved to construct diverse C-C and C-N bonds by stereoselective intercept of the in situ-formed radical intermediates. However, the Heck-type reaction of unactivated alkenes started by alkyl radical derived from intermolecular 1,5-HAT is still unknown in spite of some examples involving direct coupling of alkyl radical. As a continuation of their recent success on using N-fluoro-sulfonamides as N-radical precursor, in this manuscript, Fu, Guan and collaborators report a case of copper-catalyzed intermolecular Heck-type reaction of unactivated alkenes and N-fluoro-sulfonamides with divergent regioselectivities controlled by additive effect with broad substrate scope and good to excellent regioselectivity. With fine tuning the reaction parameters, β -lactams via a C–N reductive elimination could be readily accessed. Alkylated N-fluoro-sulfonamides enjoyed the smooth 1,5-HAT, addition to alkenes and regioselective hydride elimination under the optimized conditions. Successful application of this protocol to a wide variety of coupling partners bearing various functional groups also highlights its synthetic potential. Remarkably, the scope of radical precursors can also be extended to other readily available starting materials. On the basis of combined experimental and computational studies, a reasonable mechanism involving the critical roles of 8-aminoquinoline assisted with diverse chelating additive is also proposed for the reaction. This work provides a useful strategy for further development of other types of radical-based Heck reaction of inactivated alkenes, and will surely be of great interest to the synthetic community.

Overall, the current study is well executed, the supporting information is complete, and the results are clearly presented. This work represents a significant advance in the field of copper mediated radical coupling. In conclusion, I believe the present work fully meet the standard of this journal. Thus, I support the publication of this work in Nature Communications after addressing some minor points.

Response: We thank the Reviewer for the positive evaluation and strong support of this work.

1) This reaction showed quite a broad substrate scope with respect to the N-fluoro-sulfonamides and external and internal alkenes. Successful combination of simple alkyl group and inactivated alkene is very impressive. 8-aminoquinoline play an important role for the reaction to occur. Further explanation on this would improve the quality of this work.

Response: I agree with the Reviewer that the 8-aminoquinoline plays an important role for the success of the reaction.

The presence of 8-aminoquinoline as a bidentate auxiliary could not only simulate the intermolecular reaction as an intramolecular variant to enhance the reactivity of nonactivated olefins, but also stabilize the high-valent copper species via a [5,5]-metallabicyclic intermediate to allow for further hydride elimination or reductive elimination.

This explanation has been added in paragraph 1 of page 7 in the revised manuscript.

2) In the reaction with cycloketone oxime ester as the substrate, equal level of selectivity and efficiency was observed, clearly indicating that this reaction should be compatible with more

radical precursors, and I hope the author can explain this further.

Response: Actually, the models to alkyl radicals from *N*-fluoro-sulfonamide and cycloketone oxime ester are different; the former proceed through a SET/1,5-HAT sequence, while the latter proceed via SET/C–C fragmentation. Thus, the successful utilization of both of these two types of alkyl electrophiles in the reactions has demonstrated that after initiating the SET process of radical precursors, the subsequent generation of alkyl radicals had limited effect on the following formation of Cu(III) intermediate and hydride elimination or reductive elimination.

This explanation has been added at the end of paragraph 1 in page 15 in the revised manuscript.

3) Curious to know what would happen if privileged chiral ligands are used, such as BOX, PYBOX, BINAP etc?

Response: According to the suggestion of the Reviewer, we have tested the addition of external ligands including several chiral ligands. These ligands may compete with the 8-aminoquinoline to coordinate with the metal center, and thus influence the reaction.

As shown below, to our delight, either the vinyl product **3a** or allylic product **4a** could be obtained with high regio- and diastereoselectivities under reaction conditions A or B in the presence of these external ligands, but the reaction yields have been more or less affected. The yields of β -lactam **5a** under reaction conditions C have also decreased in varying degrees with the addition of external ligands. It should be mentioned that with chiral ligand **L3**, a low *ee* value (10%) of **5a** was observed, showing a potential asymmetric version of this reaction.

These results have been mentioned at the end of paragraph 1 in page 7 in the revised manuscript, and the details have been added in Part 2 of the revised supporting information.

Figure S1. Investigation of the formation of **3a**, **4a**, and **5a** with external ligands.

HPLC for compound *rac*-**5a** (OJ-H, *n*-hexane/*i*-PrOH = 85/15, flow rate = 0.8 mL/min, I = 254 nm, T = 40 °C) t_R = 47.9 min, 70.3 min.

HPLC for compound **5a** (10% *ee*) obtained with chair ligand **L3** (OJ-H, *n*-hexane/*i*-PrOH = 85/15, flow rate = 0.8 mL/min, I = 254 nm, T = 40 °C) t_R = 51.3 min, 74.7 min.

4) This work represents an interesting advance in the chemistry of N-radical 1,5 HAT chemistry. A quite recent review paper about this chemistry might be included. E.g., Chem. Rev. 2022, 122, 2353. A related work on 1,*n*-HAT and Heck sequence based on Pd-catalysis might also be included. Angew. Chem. Int. Ed. 2019, 58, 1794–1798.

Response: Thanks to the Reviewer's valuable suggestion. These two papers have been cited as ref. 92 and 93 in the revised manuscript.

Response to the Reviewer 2's comments

Comments:

In this work, Fu, Guan and co-workers reported a copper-catalyzed divergent regioselective Heck-type reaction of unactivated alkenes and N-fluoro-sulfonamides with a broad substrate scope (>100 examples) and good functional group tolerance. The complete control of regioselectivities was achieved by simply varying external additives, providing either vinyl products in alcohol solvent or allylic products in the presence of carboxylate and TFA. In addition, cyclization products dominated the reaction when excluding the promoting factors. Considering the importance of the Nobel-awarded Heck reaction in organic synthesis, systematic studies on the remaining challenges of this chemistry is awfully fascinating. The present work focuses on the control of regioselectivity in Heck-type reaction of unactivated alkenes, one recognized challenge as discussed in Figure 1. It is fairly impressive for a precise distinction of the several detachable β -hydrogens, and the authors have provided a satisfactory explanation based on a [5,5]-metallabicyclic intermediate through DFT calculations.

Based on the aforementioned advantages, I recommend this work to be published on Nature Communications after minor revision, and some comments are shown below:

Response: We thank the Reviewer for the positive evaluation and strong support of this work.

1) The authors stated that this is the first time for N-fluoro-amides to be applied in the coupling with alkenes. Aside from N-fluoro-sulfonamides, can other types of alkyl electrophiles be used in this divergent regioselective Heck-type reaction?

Response: Aside from N-fluoro-sulfonamides, cycloketone oxime ester could also be used in the reaction. Please see Figure 3F in the revised manuscript.

2) For **4at**, the allylic coupling of secondary alkyl radical intermediate and unactivated alkene failed. However, the reaction of tertiary alkyl radical intermediate bearing an azide group can lead to **4u**. It is better to give a brief explanation for this special case.

Response: It should be noted that the yield of **4u** produced through tertiary alkyl radical intermediate is already low (45%) compared to the allylic products bearing other functional groups, such as **4t** (-Cl, 65%), **4v** (-OBn, 65%), and **4w** (-OBz, 60%). This shows that the existence of azide group is not benefit for the allylic coupling reactions under conditions B. Therefore, for **4at** produced through less stable secondary alkyl radical intermediate, the reaction becomes worse, and no **4at** could be detected.

The comment "This may be attributed to the adverse effect of azide group on allylic coupling reactions (only 45% yield for **4u**) as well as the less stability of secondary alkyl radical intermediates" has been added in paragraph 2 of page 8 to explain this special case in the revised manuscript.

3) The β -lactam is a unique scaffold considering its bioactivity and presence in biologically relevant molecules. A representative review (Org. Biomol. Chem. 2018, 16, 6840-6852) should be cited to highlight the synthesis of β -lactams.

Response: Thanks to the Reviewer's valuable suggestion. This paper has been cited as ref. 83 in

the revised manuscript.

4) A recent DFT computational work investigated the mechanism of C–O bond activation of butanoic acid substrates bearing the 8-aminoquinoline group catalyzed by Pd(OAc)₂ (Inorg. Chem. 2021, 60, 17555-17564). Although the employed metal catalyst and substrate are different from this work, such theoretical insights into the activation of strong alkyl C(sp³) covalent bonds via the 8-aminoquinoline directing strategy should be cited.

Response: Thanks to the Reviewer's valuable suggestion. This paper has been cited as ref. 98 in the revised manuscript.

5) Imaginary frequencies of the transition states on the PES should be provided in Figure 6.

Response: According to the suggestion of the Reviewer, all imaginary frequencies of the transition states have been provided in revised Figures 5 and 6.

Response to the Reviewer 3's comments

In this manuscript, Fu, Guan, and coworkers developed the divergent regioselective intermolecular Heck-type reaction via a directed copper-catalyzed coupling of N-fluoro-sulfonamides and unactivated alkenes. Recently, they have reported the intermolecular Heck-type reactions of unactivated alkenes with alkyl electrophiles, including α -bromo esters in 2018 (ref. 34) and cycloketone oxime esters in 2020 (ref. 41). In this work, they extended this chemistry to N-fluoro-sulfonamides, and successfully realized divergent regioselective Heck-type reaction. As we know, it's difficult to adjust the divergent regioselectivities from a highly reactive Cu(III) intermediate, and the β -lactams are commonly obtained (ref. 43-47). The Heck-type vinyl coupling products are only observed in special cases.

Herein, the authors realized the complete regioselectivity by simply varying the external additives. Control experiments and DFT calculations illustrated a detailed reaction profile of the key Cu(III) intermediate, highlighting the distinct effect of the external additives. Moreover, this protocol shows a broad substrate scope for both alkenes and structurally diverse N-fluoro-sulfonamides, producing the corresponding products with excellent regio- and stereoselectivities. More than 150 examples are provided, showing the broad scope, as well as some limitations of the substrates. And the manuscript and SI is well organized. Overall, this reviewer supports the acceptance of the manuscript after a minor revision.

Response: We thank the Reviewer for the positive evaluation and strong support of this work.

1. Have the authors tried the γ - δ unsaturated amide substrates, for the three kinds of reaction pathways?

Response: According to the suggestion of the Reviewer, we have tried to test the reactions of γ - δ unsaturated 8-aminoquinoline amide with N-fluoro-sulfonamides **2a**. Unfortunately, all of the reactions failed under conditions A-C.

These results have been added at the bottom of Table 1, and mentioned in paragraph 1 of page 7 in the revised manuscript.

2. Does the authors have tried some other types of transition metals, such as the earth-abundant nickel or cobalt salt?

Response: We have tried to replace the copper catalyst with catalytic amount of Ni(cod)₂ or Co(acac)₂. However, no desired product could be detected and the majority of alkene substrate was recovered.

These results have been added in Table 1 as entries 16 and 17, and briefly mentioned at the end of paragraph 1 in page 5 in the revised manuscript.

3. For the control experiment in Figure 3F, C–N reductive elimination reaction by using **1a** and cycloketone oxime ester as substrates was suggested to be added.

Response: According to the suggestion of the Reviewer, we carried out the reaction of **1a** and cycloketone oxime ester under conditions C. The desired β -lactam **20** was obtained in 68% yield via C–N reductive elimination.

This data has been added in Figure 3F in the revised manuscript.

¹H NMR Spectra of **20**

¹³C NMR Spectra of 20

4. In the proposed mechanisms (Figure 4), the reaction was initiated by Cu(I) species, rather than the added Cu(II) catalyst. Please explain why. Is there any rationale or reference?

Response: The disproportionation reaction of Cu(II) salt would generate active Cu(I) species and Cu(III), which has also been proposed by You in 8-aminoquinoline-directed oxidative C–H cross-coupling (*Org. Lett.* **2014**, *16*, 2884-2887; cited as ref. 87).

The statement “the disproportionation reaction of Cu(II) catalyst generated a Cu(I) species” has been added at the beginning of the “Proposed reaction mechanism” section in the revised manuscript.

5. The large-scale experiment should be added in the manuscript and SI.

Response: According to the valuable suggestion of the Reviewer, the large-scale experiments of alkenyl amide **1a** and *N*-fluoro-sulfonamide **2a** under conditions A-C (see Page 104, 146, and 174) have been added in the revised supporting information.

6. Line 117: change (r.r. < 20:1) to (r.r. > 20:1);

Response: We have made the revision according to the suggestion of the Reviewer.

In SI, Page S286: the signal-to-noise for ¹³C NMR of compound **3at** is poor. Please check it.

Response: We have replaced both of the ¹H and ¹³C NMR of compound **3at** with new ones in the revised supporting information.

¹H NMR Spectra of 3at

¹³C NMR Spectra of 3at

REVIEWERS' COMMENTS

Reviewer #1 (Remarks to the Author):

The author satisfactorily answered the reviewer's questions, the overall article has been further improved, and the article is now ready for acceptance and publication.

Reviewer #2 (Remarks to the Author):

The authors have revised the manuscript according to my review comments, it could be accepted now.

Reviewer #3 (Remarks to the Author):

In their revised manuscript and comments, the authors have addressed the main points raised by the reviewers to a good degree. Although it is a pity that the γ - δ unsaturated amide substrates were not suitable for the reaction at the present stage, this does not affect the novelty and integrity of the article. In fact, the present Heck-type reaction worked well with a variety of unactivated alkenes, affording the desired products with excellent regio- and stereoselectivities. Therefore, the reviewer would now support the publication of this paper in Nature Communications.

Response to the Reviewer 1's comments

Reviewer #1 (Remarks to the Author):

The author satisfactorily answered the reviewer's questions, the overall article has been further improved, and the article is now ready for acceptance and publication.

Response: We thank the Reviewer for the positive evaluation and strong support of this work.

Response to the Reviewer 2's comments

Reviewer #2 (Remarks to the Author):

The authors have revised the manuscript according to my review comments, it could be accepted now.

Response: We thank the Reviewer for the positive evaluation and strong support of this work.

Response to the Reviewer 3's comments

Reviewer #3 (Remarks to the Author):

In their revised manuscript and comments, the authors have addressed the main points raised by the reviewers to a good degree. Although it is a pity that the γ - δ unsaturated amide substrates were not suitable for the reaction at the present stage, this does not affect the novelty and integrity of the article. In fact, the present Heck-type reaction worked well with a variety of unactivated alkenes, affording the desired products with excellent regio- and stereoselectivities. Therefore, the reviewer would now support the publication of this paper in Nature Communications.

Response: We thank the Reviewer for the positive evaluation and strong support of this work.